# Deep Reinforcement Learning for Indoor Mobile Robot Path Planning

**DOI:** 10.3390/s20195493

**Published:** 2020-09-25

**Authors:** Junli Gao, Weijie Ye, Jing Guo, Zhongjuan Li

**Affiliations:** School of Automation, Guangdong University of Technology, Guangzhou 510006, China; jonygao621@gdut.edu.cn (J.G.); ye_wei_jie@163.com (W.Y.); lizhongjuan@126.com (Z.L.)

**Keywords:** mobile robot, reward function, path planning, incremental training mode, deep reinforcement learning, deep neural network, generalization

## Abstract

This paper proposes a novel incremental training mode to address the problem of Deep Reinforcement Learning (DRL) based path planning for a mobile robot. Firstly, we evaluate the related graphic search algorithms and Reinforcement Learning (RL) algorithms in a lightweight 2D environment. Then, we design the algorithm based on DRL, including observation states, reward function, network structure as well as parameters optimization, in a 2D environment to circumvent the time-consuming works for a 3D environment. We transfer the designed algorithm to a simple 3D environment for retraining to obtain the converged network parameters, including the weights and biases of deep neural network (DNN), etc. Using these parameters as initial values, we continue to train the model in a complex 3D environment. To improve the generalization of the model in different scenes, we propose to combine the DRL algorithm Twin Delayed Deep Deterministic policy gradients (TD3) with the traditional global path planning algorithm Probabilistic Roadmap (PRM) as a novel path planner (PRM+TD3). Experimental results show that the incremental training mode can notably improve the development efficiency. Moreover, the PRM+TD3 path planner can effectively improve the generalization of the model.

## 1. Introduction

A mobile robot with autonomous navigation capability can search for one path within specific time to move from a start point to an endpoint without collision [1]. For indoor environments, traditional navigation frameworks employ Simultaneous Localization and Mapping (SLAM) [2] to map unknown environments, then exploit localization algorithms, for instance, Adaptive Monte Carlo Localization (AMCL) [3] to determine the current location of a robot and move it to the destination through a path planning module [4,5]. Path planning is one of the key technologies in navigation system, which usually includes the global as well as local planners [6,7]. The global planner searches the global map to obtain a feasible path by means of planning algorithms, such as Dijkstra, A-star (A*), PRM, Rapidly Exploring Random Tree (RRT) algorithms, and so on [8,9,10,11,12]. The Dijkstra is a classic algorithm for finding the shortest path between two points due to its optimal capability for global planning. The A* achieves better performance by using heuristics to guide its search, which can be seen as an extension of Dijkstra. A motion planning algorithm sampling-based, PRM, solves the problem of determining a path between a starting configuration and a goal configuration while avoiding collisions. The RRT is an algorithm designed to search non-convex, high-dimensional spaces by randomly building a space-filling tree. However, the local planner is used for real-time collision-free based on the global path by means of planning algorithms, such as Artificial Potential Field (APF), Dynamic Window Approach (DWA) and Timed Elastic Bands (TEB) [13,14,15]. The APF treats a robot’s configuration as a point in a potential field that combines attraction to the goal, and repulsion from obstacles to track the safe path. One velocity-based local planner, DWA, calculates the optimal collision-free velocity for a robot required to reach its goal, which translates a Cartesian goal (x,y) into a velocity (v,w) command. The TEB merges the states, control inputs and time intervals into a joint trajectory to enable the path planning of time-optimal trajectories. Most of the global path planning algorithms are suitable for path-finding of static scenes and require prior map information. With the increase of complexity and uncertainty of the environment, their efficiency will be greatly reduced. The local path planning obtains the surrounding environment information through sensors, and generates a collision-free path in a related small scope with better adaptability and embarrassed local minimum. In conclusion, the traditional path planning suffers from the issues of map-dependence, low real-time performance, or expert experience-dependence [16].

In recent years, there is a growing trend to apply DRL for mobile robot indoor path planning with much success achieved [17]. DRL has the advantages on map-free, strong learning ability and low sensor accuracy dependence, etc., whereas the prohibitively long training duration for DRL-based path planning has severely hindered its wide application to mobile robots, especially for the scenario of limited computation resources. In most cases, DRL-enabled robots need to be trained in a 3D simulation environment with a physics engine. In this scenario, the robot’s movement is limited by physical rules. It is implemented based on interactive trial-and-error, which dramatically increases the training duration [18]. When the environments or tasks are complex, it is difficult for the algorithm to converge to the desired targets with randomly initialized network parameters. For example, the weights and biases [19] may further deteriorate the performance of the DRL-enabled path planning. Moreover, because actions are always carried out whenever influences from observation states are derived in most DRL framework and they are mostly local, this may reduce the ability of a mobile robot to make a satisfactory global decision.

This paper aims to tackle the above issues, and the major contributions are summarized as follows:We propose an incremental training mode, which includes environment and network parameters transfer to improve the development efficiency and convergence of the DRL-based path planning for a mobile robot. It can also contribute to test and evaluate the algorithms, network structure, and so on, to polish the path planning scheme conveniently.We enhance the PRM path planning approach with TD3 to be a PRM+TD3 planner, which outperforms the popular algorithms, for instance, A*+DWA, A*+TEB and TD3, for a mobile robot indoor navigation.We extensively evaluate and analyze the performance of PRM + TD3 on various application scenes, compared with A*+DWA, A*+TEB and TD3 according to the same environment and criteria.

The rest of this paper is organized as follows. Section 2 introduces the related works. Section 3 presents the path planning problem on a mobile robot, and details the proposed incremental training mode. Then, the related graphic search algorithms and RL algorithms are tested and evaluated. Section 4 formulates the observation states and reward function. Afterwards, build the environment and agent for DRL. Section 5 analyzes the experimental results before conclusion in Section 6.

## 2. Related Work

DRL was first applied in discrete control of mobile robots to realize obstacle avoidance [20]. Pfeiffer et al. [21] adopted the planning results from Dijkastra and DWA as the labels for convolutional neural networks (CNN) to train a path planner. Tai et al. [22] proposed the end-to-end obstacle avoidance strategy of mobile robot based on deep Q learning. After that, Tai et al. [23] presented a map-free indoor navigation system according to sparse laser radar signals and DRL. Faust et al. [24] achieved long-range robotic navigation by combining PRM and RL. Francis et al. [25] used PRM as the sampling-based planner, and AutoRL as the RL method in the indoor navigation context to realize long-range indoor navigation. Zeng et al. [26] presented a jump point search improved asynchronous advantage Actor-Critic (JPS-IA3C) for robot navigation in dynamic environments, which computes an abstract path of neighboring jump points (sub goals) by the global planner JPS+ and learns the control policies of the robots’ local motion by the improved A3C (IA3C) algorithm. Here, the IA3C denotes the local planner combined with the global planner to realize path planner for a robot. Iyer et al. [27] proposed an approach of collision avoidance robotics via meta-learning (CARML) to explore a 2D vehicle navigation equipped with a lidar sensor and compare against a baseline TD3 solution to solve the same problem. Sánchez-López et al. [28] presented a real-time 3D path planning solution for multi-rotor aerial robots, where a probabilistic graph is utilized to sample the admissible space without taking into account the existing obstacles and the generated probabilistic graph is then explored by an A* discrete search algorithm with an artificial field map as cost function in order to obtain a raw optimal collision-free path. Kim et al. [29] designed a motion planning algorithm using TD3 with hindsight experience replay to enhance sample efficiency, where the designed paths are smoother and shorter than those designed by PRM. Zhu et al. [30] developed a vision-based indoor navigation system in a data-driven manner. Stooke et al. [31] accelerated training speed by improving the utilization of CPU and GPU based on optimized RL algorithms. Carlos et al. [32] decomposed complex problems into multiple sub-problems to search the maze path according to dynamic programming. Chiang et al. [33] considered PRM and RRT as global path planning methods, respectively, to find intermediate way-points for DRL algorithm in large-scale indoor navigation.

Inspired by the above related works, we propose a novel incremental training mode for path planning of DRL-enabled mobile robot, to reduce the time for algorithm training and testing. Through transferring the network parameters from simple to complex environment, the convergence issue caused by random initialization is alleviated. Moreover, a path planner of PRM+TD3, which combines the global path planning algorithm PRM and DRL algorithm TD3 [34] is proposed to decompose large-scale paths into multiple local paths and improve the generalization of trained models. Experimental results show that the incremental training mode can significantly improve the development efficiency. Besides, PRM+TD3 path planner has better generalization in large-scale scene.

## 3. The Methodology

At first, we present the path planning problem on mobile robots. Next, detail the proposed incremental training mode. After that, the related graphic search algorithms and RL algorithms are tested and evaluated for the next work on DRL-based path planning.

### 3.1. Problem Description

In this paper, we formulate the path planning problem as a Markov Decision Process (MDP), modeled as a tuple (S,A,T,R,γ) [35], where *S* and *A* denote the system’s state space and action space, respectively. At each time step, the system takes an action a∈A and moves from the current state s∈S to a new state $s_{{}^{\prime }}\in S$. $T\left(s,a,s_{{}^{\prime }}\right)=P\left(s_{{}^{\prime }}|s,a\right)$ gives the transition probability that the system lies in $s_{{}^{\prime }}$, after taking action *a* in state *s*. R:S×A→R is the reward function. The system receives a real-valued reward R(s,a) that depends on its state *s* and action *a* at each time step. γ∈[0,1] is a discount factor that reflects the preference of immediate rewards over future ones. The agent’s action selection is modeled as a map called policyπ:S→A prescribes an action a∈A for each state s∈S. A policy is called optimal if it achieves the best expected return from any initial state. The value function $v_{\pi }\left(s\right)$ is defined as the expected return starting with state *s*, i.e., S=s, and successively following policy
π, which estimates "how good" it is to be in a given state. While the maximum possible value of $v_{\pi }\left(s\right)$ is defined as $v_{*}\left(s\right)$. Given a state *s*, an action *a* and a policy
π, the action value of the pair (s,a) under π is defined as $q_{\pi }\left(s,a\right)$, whilst the optimal action value function is called $q_{*}\left(s,a\right)$.

Under a model-based environment, i.e., the state transition probability *P* is known, the MDP problems can be solved by dynamic programming techniques. As for model-free, the model can be fitted by means of sampling methods, which are suitable for RL. The main difference between the classical dynamic programming methods and RL algorithms is that the latter do not assume knowledge of an exact mathematical model of the MDP and they target large MDPs where exact methods become infeasible. The expectations are approximated by averaging over samples and using function approximation techniques to cope with the need to represent value functions over large state-action spaces for model-free RL. The two basic approaches to compute the optimal action value function are value iteration and policy iteration. An alternative method is to search directly in the policy space, in which the problem becomes a case of stochastic optimization, for instance, the gradient-based and gradient-free methods. We adopt RL algorithm TD3 for robot path planning, which utilizes the optimal value function and policy function detailed in Section 4.4.

### 3.2. Incremental Training Mode

The proposed incremental training mode includes environment/network structure/parameter transfer, and the training process. Here, the word “incremental” means the training process and application scene are more and more complex based on the previous phase, successively.

#### 3.2.1. 2D to 3D Environment Transfer

Usually, the DRL model for a robot can be prior-trained in a simulation environment equipped with a physics engine before real application. We take the Gazebo in Robot Operation System (ROS) [36], a widely used robot platform, as the 3D simulation environment, which integrates with the physical Open Dynamics Engine (ODE). Furthermore, the mobile robot TurtleBot3 is modeled by Unified Robot Description Format (URDF). As shown in Figure 1a, TurtleBot3 implements path planning for the preset targets to avoid the rotating obstacles in this environment. The tasks, such as DRL algorithm verification, parameter debugging, inevitably consume significant computational time and resources in a 3D simulation environment loaded with a physics engine, which leads to inefficiencies.

To improve the development efficiency, we developed a corresponding 2D Gym environment without physics engine as shown in Figure 1b, according to the Pyglet package of Gym [37]. The basic characteristics in Gym are as same as Gazebo in Figure 1a, i.e., continuous space, real-time obstacle avoidance, preset goals, perceptible environment, and so on. All the positions of robot and rotating obstacles, are calculated, i.e., known, based on the velocity and time of the previous state, which correspond to the lidar readings in Gazebo. Furthermore, Gym has the interface for Gazebo, which means the algorithm codes in Gym can be reused in Gazebo easily, except the codes about environments. The general principle of the incremental training mode is to deploy the complex tasks such as DRL algorithm verification and parameter debugging to the lightweight Gym for preliminary training, before transferring the best solution back to Gazebo. Experiments have demonstrated that the proposed solution is capable of a 5-fold increase in development efficiency for the DRL-based path planning. As an illustration, it takes about 26 and 5 h in Gazebo and Gym, respectively, with the same tasks and hardware platform [38].

The position information of a mobile robot is reported in polar coordinates, the reward function is defined in Equation (Equation 2), and TD3 is adopted as the DRL algorithm. In order to compare the convergence and generalization capability of the DRL algorithm in a complex environment, we build the 8 × 8 m^2^ and 13 × 13 m^2^ simulation environment as shown in Figure 2, where the Figure 2a,b are used for training and testing, respectively. The corresponding experimental results and analyses are detailed in Section 5.

**Remark** **1.**
*The network structure (including parameters) and reward function in Gym are as same as Gazebo, except the state set S, which also is the basis for “2D to 3D transfer”.*


#### 3.2.2. Parameter Transfer

DNN in DRL are sensitive to the initial weight assignments. Random initialization of these parameters may cause slow convergence or even non-convergence at times. Therefore, taking TD3 as an example, before formal model training, we first train $1\times 10^{5}$ time steps in a 4 × 4 m^2^ simple scene, then transfer the corresponding model parameters to a 8 × 8 m^2^ complex scene. The parameters obtained after $1\times 10^{5}$ time steps in the previous phase are adopted as the initial weights. As shown in Figure 3, the proposed transfer approach yields better convergence on TD3, compared with training $2\times 10^{5}$ time steps directly (i.e., no_transfer) in the 8 × 8 m^2^ scene, which cannot reach the target point. Moreover, the reward tends to be −100 instead of a positive value, which indicates that the model has failed to learn from the data to perform the desired path planning task.

#### 3.2.3. The Training Process

The incremental training process for DRL is revealed in Figure 4, where we take TD3 and PRM+TD3 as examples. Firstly, the time-consuming works are conducted in the lightweight 2D Gym, for instance, setting reward function, evaluating algorithms on path planning, debugging parameters, and so on. Secondly, transfer the optimal solution to the 3D Gazebo (4 × 4 m^2^) including network weights *W*, biases *b*, algorithm codes, etc. and keep training for $1\times 10^{5}$ time steps. Thirdly, after incremental transfer again, the model is further trained for $5\times 10^{5}$ time steps in an 8 × 8 m^2^ 3D scene. Since then, we can further evaluate the generalization of DRL algorithms following Figure 4.

### 3.3. Graphic Search Algorithms

PRM and RRT are the classic sampling-based path planning algorithms, which belong to the graphic search techniques. The simulation results, using PRM and RRT algorithms under the 2D Gym environment, namely, graphic search based on sampling to quickly select intermediate points from the global map, are shown in Figure 5. The green, yellow and red points denote the starting, ending and sampled points, respectively, and the red curve is named global path. It is noted that the fewer intermediate way-points, the more stable the paths can be generated while maintaining the connectedness between the start point and endpoint. To ensure the performance of the RRT, the step size generally keeps very small to cause denser way-points on the map (Figure 5a. In contrast, the PRM only selects the way-points from the limited sampling points, which is preferred for our path planning tasks (Figure 5b).

### 3.4. RL Algorithms

To determine the optimal RL algorithm for path planning, we evaluate several popular algorithms for continuous state-action space, like Proximal Policy Optimization (PPO) [39], Deep Deterministic Policy Gradient (DDPG) [40], Soft Actor-Critic (SAC) [41] and TD3 [34] based on 2D Gym, which can be reused in 3D Gazebo easily, discussed in Section 3.2.1. The DDPG, SAC and TD3 fuse value and policy function, while the PPO is only based on policy gradient. All the algorithms were implemented by Tensorflow 2.0, running $5\times 10^{5}$ time steps. The observation states and reward function follow the Equations (Equation 1) and (Equation 2), respectively. Take the negative of the distance between agent and target position as the instant reward. The collision-free reward is 100, and the negative reward is −150 with collision. As shown in Figure 6, the convergence of all the RL algorithms are depicted. By comparison, TD3 has better convergence, which means the mobile robot can learn the planning ability to reach the target position without collision much more quickly.

A desirable DRL algorithm can output an instant reward and action instruction based on the observation values. For mobile robots, however, the observations by the lidar or vision camera are mostly local, which may limit to making globally optimal decisions. On the other hand, it is difficult to obtain observations in a larger scene for global path planning on the fly, where the final target needs to be broken into a series of intermediate way-points. Furthermore, the DRL algorithm performs multiple path planning based on these way-points, for long-range navigation in this scene. The algorithm for selecting intermediate way-points should satisfy that any two points including the start point and endpoint must be able to be connected. And the connection should not pass through the obstacles.

After a systematic consideration about the aforementioned methodology, we propose the PRM+TD3 path planner, which utilizes global planning algorithm PRM to select intermediate way-points, and integrates with TD3 algorithm for local planning.

**Remark** **2.**
*The convergence of the PRM+TD3 algorithm is only determined by TD3, because the PRM only supplies the way-points for TD3 with no impact on the convergence, verified through the next experiments.*


## 4. Deep Reinforcement Learning

In this section, we formulate the observation states and reward function, then build the DRL environment and the agent for the next experiment and analysis in Section 5.

### 4.1. Observation States

The training goal of the DRL model is to maximize the total reward *R* during interactions between the agent and environment. At a given time step *t*, the state $s_{t}$ is composed of the lidar data $o_{m}$ (10-dimensions), the geometric relationship, i.e., the angle and distance $o_{g}$ (2-dimensions) between the agent and target point, the linear velocity $v_{t-1}$ and angular velocity $\omega_{t-1}$ (2-dimensions) at the previous time step. Regarding the DRL model as a planner f(*), it maps the observation states $s_{t}$ as the linear velocity $v_{t}$ and angular velocity $\omega_{t}$ (2-dimensions, Action set *A*, the DNN’s outputs) of a mobile robot at the current time step, illustrated in the Equation (Equation 1).
(1)$$v_{t},\omega_{t}=f\left(s_{t}\right)=f\left(o_{m},o_{g},v_{t-1},\omega_{t-1}\right)$$

Whenever the observation states $s_{t}$ change, the environment will feed back an instant reward $r_{t}$, which refers to the correctness of the current decision. The more it meets the task requirements, the greater reward $r_{t}$ will be assigned, and vice versa. Due to lack of the prior knowledge of the environment, the transition probability *P* between each state is unknown, so model-free DRL algorithms are preferred.

### 4.2. Reward Function

The observation states aforementioned in Section 4.1 determine the input/output of the DRL algorithm. Next, we design the reward function to supervise the agent to learn and obtain the optimal policy. The instant reward is a key element in DRL, which can be obtained by the reward function at each time step. Therefore, the model can learn the specified functions on condition that the reward function is designed reasonably. For robot path planning, the learning goal is that a robot quickly moves from a start point to an endpoint without collision. According to the literature [23], the reward function is defined in the Equation (Equation 2), beyond that we augment an inflation layer for more reliable obstacle avoidance.
(2)$$r\left(s_{t},a_{t}\right)=\left\{\begin{array}{ccc} 200 & \; & if\;(d_{t}<d_{goal})\\ -150 & \; & if\;(d_{t}<d_{min})\\ 1-\frac{d_{t}}{r_{i}+r_{s}} & \; & if\;(d_{t}<r_{i}+r_{s})\\ \alpha *(d_{goal}-d_{t}) & \; & otherwise \end{array}\right.$$
where $d_{t}$ represents the distance between the current position of a robot and target point, and $d_{min}$ is the defined minimum distance from the obstacle. We take $r_{i}+r_{s}$ as the inflation layer to maintain more safety from obstacles, including the radius of a robot $r_{i}$ and the safe distance $r_{s}$. $d_{goal}-d_{t}$ denotes whose opposite value between the current position and target point as an instant reward. The threshold distance $d_{goal}$ determines whether it can reach the target point. α is a scale factor. In view of the definition, a mobile robot will move from the start point to target point while maintaining a certain safe distance from obstacles based on the Equation (Equation 2).

### 4.3. DRL Environment

To integrate robot path planning with DRL, it needs to transit the planning problems into the framework of DRL. Therefore, We design the agent/environment interface according to the Equation (Equation 3). The main part of DRL environment is consisted of the functions step(), reset() and render(), where the reset() and render() are often embedded in the step(). The step() determines the interactive rules between the agent and environment and implements the path planning algorithm. While the reset() and render() are used for the DRL environment reset and visualization.
(3)Env=(step,reset,render)

As shown in Figure 7, the agent interacts with the Gym or Gazebo environment through the step(). After initialization, the agent selects the action *a* to interact with the environment. When it changes, the agent determines whether has collisions or up to the maximum time step by the Boolean ending flag done. If Yes, it resets the environment, and returns the current reward $r^{*}$, next state $s^{{}^{\prime }}$ and ending flag done. For No, provided that the agent has moved to the target position, it regenerates another target position and return $\left(r^{{}^{\prime }},s^{{}^{\prime }},done\right)$. If still not reach, it returns $\left(r^{{}^{\prime \prime }},s^{{}^{\prime }},done\right)$ directly. Here, the $r^{*}$/$r^{{}^{\prime }}$/$r^{{}^{\prime \prime }}$ are the reward $r_{t}$ under different conditions. It loops until up to the maximum time step, while the data $\left(s,a,r,s^{{}^{\prime }},done\right)$ is used for DRL algorithm training.

### 4.4. The Agent

DNN can be used for nonlinear approximation of the value and policy functions in DRL. The network framework of TD3 [34,42] adopted in this paper is shown in Figure 8, including one value function network (two-Q) and one policy function network, in where each layer is consisted of connection type (fully connected: FC), neurons and activation function except the input layer. TD3 fuses the value and policy functions, which adopt the similar DNN.

After determined the network framework, the approximation approaches to train the DNN is presented as the Equations (Equation 4)–(Equation 6). The Equation (Equation 4) is adopted by the two Q networks to update the value function and use the minimum Q value based on Bellman equation to descend the overestimation problem. The Equation (Equation 5) updates the Q function by gradient descent to decline the approximation error, while the Equation (Equation 6) updates the policy by gradient ascent. The rate for the Q function and policy function is 2:1, i.e., delayed update policy to relieve the fluctuations during training. For the policy function network, Gaussian noise ε is added before output, which contributes to the exploration ability during the interactions between the agent and environment.
(4)$$y\left(r,s^{\prime },done\right)=r+\gamma \left(1-done\right)\min_{i=1,2}Q_{\phi_{target_{i}}}\left(s^{\prime },a^{\prime }\left(s^{\prime }\right)\right)$$
(5)$$\nabla_{\phi_{i}}\frac{1}{\left|B\right|}\sum_{(s,a,r,s^{\prime },d)\in B}{\left(Q_{\phi_{i}}\left(s,a\right)-y\left(r,s^{\prime },done\right)\right)}^{2},\;for\;i=1,2$$
(6)$$\nabla_{\theta }\frac{1}{\left|B\right|}\sum_{s\in B}Q_{\phi_{1}}\left(s,\mu_{\theta }\left(s\right)\right)$$
where *y* is the target value for the two Q function. *r* and $s^{{}^{\prime }}$ denote the reward and next state, respectively. The Boolean ending flag done indicates the iteration is over or not at the batch size *B*. The discount factor is γ∈[0,1]. While, $\phi_{{target}_{i}}$ and $\phi_{i}$(i=1,2) are the Q function parameters. The target action after clipped is $a^{{}^{\prime }}\left(s^{{}^{\prime }}\right)$. θ is the policy parameter and $\mu_{\theta }\left(s\right)$ is the policy function. For our implementation of TD3 [34], we use the codes provided by the author (https://github.com/sfujim/TD3).

## 5. Experiment and Analysis

Given the experiment settings and evaluation criteria first. Then, test and evaluate the proposed PRM+TD3 path planner with the A*+DWA, A*+TEB, TD3 based on the same criteria and environment. Analysis the path planning results and generalization at different application scenes in the end.

### 5.1. Experiment Settings

For the application of path planning on the mobile robot TurtleBot3, the algorithms to be compared and analyzed are divided into three groups: I-A*+DWA, II-A*+TEB, and III- PRM+TD3, respectively. Here, I/II are the traditional path planning algorithms, III is the combination of the traditional global path planning algorithm PRM and modern DRL TD3. In contrast with the path planning and generalization ability of these algorithms, the application scene was classified into small-scale (8 × 8 m^2^) and large-scale (13 × 13 m^2^), relatively. Both scenes are Gazebo-based. All experiments are conducted on a standard laptop with I5-8250U@1.6 GHz processor and 8 GB RAM. The parameters of TurtleBot3 are set as follows: linear velocity [0, 0.2] (m/s), angular velocity [−2, 2] (rad/s), scanning distance of 2D lidar [0.2, 3.5] (m), and scanning angle [0, 2] (rad). The evaluation criteria are explained in Table 1.

### 5.2. Small-Scale Scene

To verify the planning performance of DRL in different scenes, we implement TD3 in a small-scale scene (8 × 8 m^2^) first, then compare TD3 with PRM+TD3 in large-scale scene (13 × 13 m^2^) (Section 5.3). At the beginning, we generate 7 target points in sequence to conduct path planning with A*+DWA, A*+TEB, and TD3, respectively, while four obstacles rotate anticlockwise. The simulation results are plotted in Figure 9, where the red/purple/green curve represent the planned path with the corresponding I/II/III algorithms, respectively. It can be seen that TD3 is better than the other two.

From Table 2, TD3 does not need prior map in comparison with I/II algorithms. The traditional path planning requires global-planning by the map first, then the local-planning is performed after the local cost map is generated by the sensor data. That is why the $t_{step}$ of I/II algorithms can only have a variable range with dynamic environment. The more complex the local cost map, the less efficient of the planning. As for TD3, the decision at each step is generated through a fixed forward neural network operation, whose duration maintains at millisecond level, i.e., better responsiveness to complex environments. In a dynamic and narrow environment, the traditional algorithm I/II occasionally have collision issues due to its inferior real-time performance. While TD3 can almost complete the planning task. With the inherent mechanical characteristics, the collision seldom occurs at the starting phase, when the agent and obstacles are very close. Because of the embedded physical engine ODE in Gazebo, the acceleration and velocity of the agent are smaller than the mobile obstacles, it is consistent with the practice and the collision will disappear at the normal running phase.

### 5.3. Large-Scale Scene

In the 13 × 13 m^2^ scene, the group III still only uses TD3 at first. The planning effect of the algorithms are verified through a long-distance target, shown in Figure 10. It can be found that TD3 (Figure 10c) cannot complete the task without the global path planning to provide intermediate way-points, while the traditional I/II algorithms can achieve it (Figure 10a,b). It is proved that although TD3 has higher real-time performance, it is incapable of global planning, particularly in large-scale scene.

Next, the PRM is added to the III group, which provides the global intermediate way-points for TD3, and decomposes the path into multiple local sub-paths, as shown in Figure 11a. It samples 50 points in the map first, then performs path searching. The red curve represents the global path planned by PRM. TD3 takes the intermediate way-points from the global path as the sub-targets, and performs path planning in multiple segments to achieve large-scale path, plotted in Figure 11b.

From Table 3, it can be seen that in large-scale path planning, TD3 requires PRM to search for intermediate way-points in the global map, i.e., map dependence ($d_{ep}=yes$). Since PRM is complete probabilistic, the results of each planning are different and might not be optimal. The path length $l_{path}$ and total time consumption $t_{path}$ for PRM+TD3 are longer than the other two. Nevertheless, the single-step time $t_{step}$ for PRM+TD3 is still shortest, which has better flexibility in dynamic and narrow small-scale scenes. As listed in Table 3, the 13 × 13 m^2^ scene has no dynamic narrow area compared with the 8 × 8 m^2^ scene, so the percentage completed is higher than the other algorithms. At the same time, because PRM can provide intermediate way-points to enable TD3 to have better generalization capabilities in the 13 × 13 m^2^ testing scene, which can fulfill the path planning.

## 6. Conclusions

Aiming at addressing the issues of low development efficiency, poor convergence, and weak generalization ability of DRL-enabled robot in a 3D simulation environment, a incremental training mode is proposed for conducting the path planning. Firstly, we deploy the complicated tasks such as DRL algorithm verification and parameter debugging in a lightweight 2D Gym environment for training, then transfer the best solution obtained from Gym to the 3D Gazebo environment, which can enhance the development efficiency of the DRL-enabled mobile robot by 5 times for path planning. By the network parameters transferring, we perform prior training in a 4 × 4 m^2^ simple scene first, then transfer the model to an 8 × 8 m^2^ scene to re-train, which can alleviate the convergence issue of TD3 caused by the random initialization of DNN. Given that TD3 only fits well in small-scale scene, PRM+TD3 path planner is proposed for larger scene. Experiments show that it can decompose long-distance target navigation into multiple sub-target, and get better generalization to new large-scale scene. Even if the incremental training mode can improve the development efficiency, convergence and generalization of the DRL enabled mobile robot, the PRM is a type of sampling algorithm in nature. The results of each planning are different and might not be optimal. Our further work will improve this kind of non-optimal path planning algorithm with DRL [26] continuously. Moreover, we will also consider the path planning for multiple robots [43,44].

## Figures and Tables

**Figure 1 sensors-20-05493-f001:**
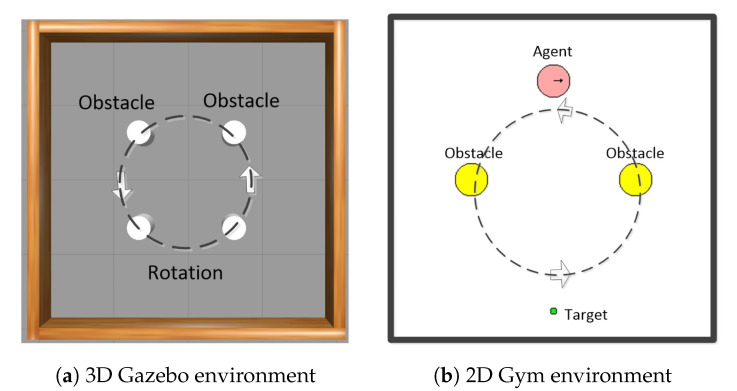
The 2D and 3D simulation environments (4 m × m^2^).

**Figure 2 sensors-20-05493-f002:**
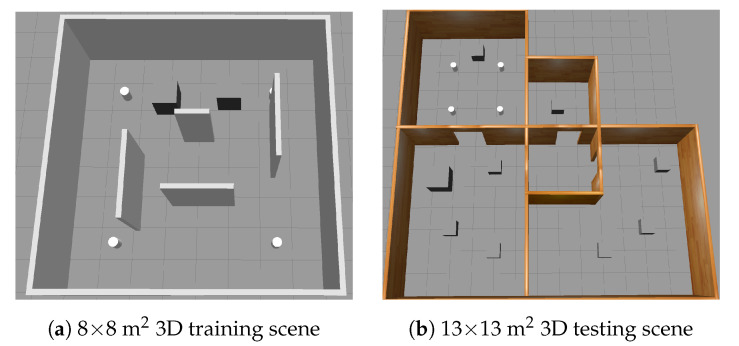
Complex 3D simulation environments.

**Figure 3 sensors-20-05493-f003:**
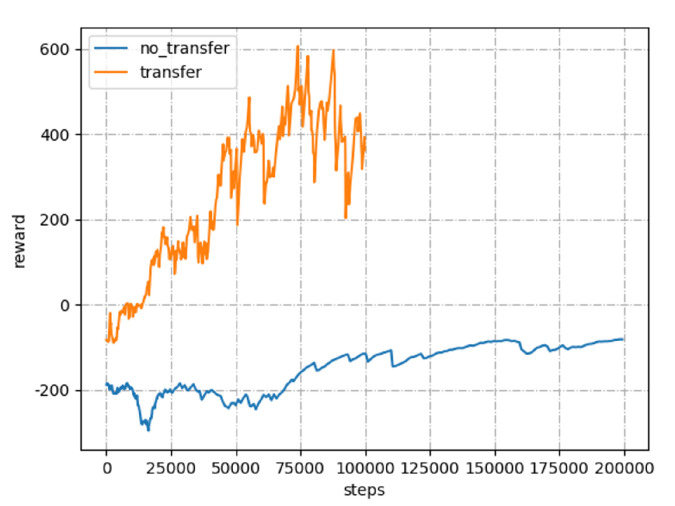
Parameters transfer effect.

**Figure 4 sensors-20-05493-f004:**
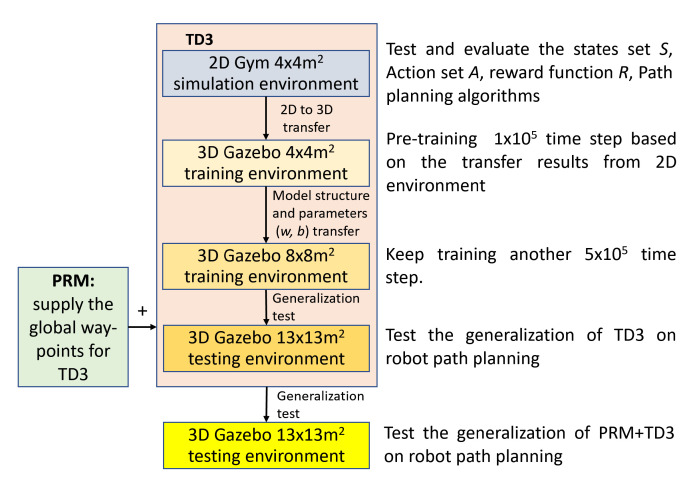
The incremental training process.

**Figure 5 sensors-20-05493-f005:**
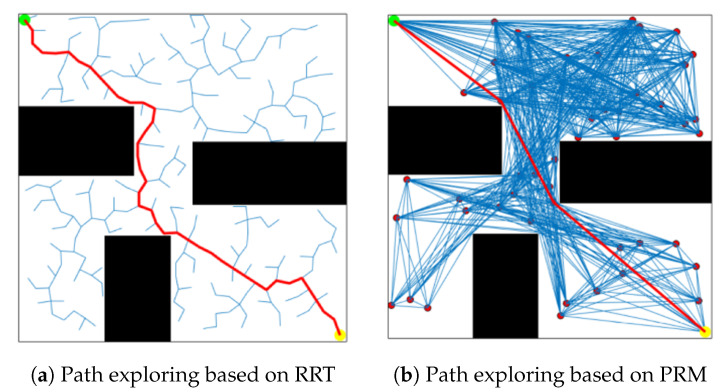
Graph search algorithm based on sampling.

**Figure 6 sensors-20-05493-f006:**
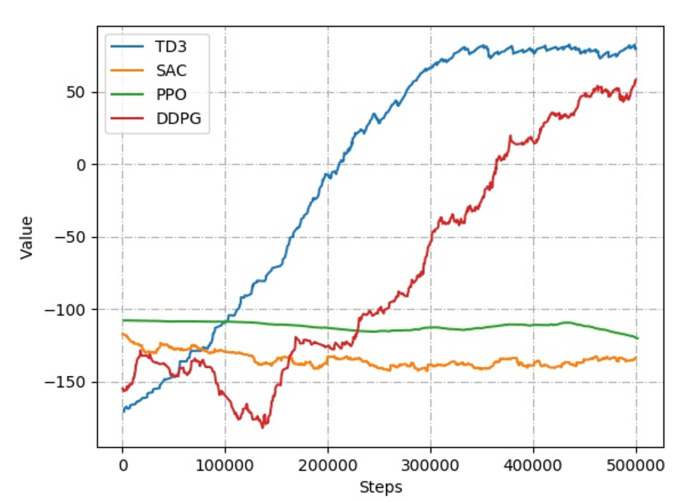
The convergence evaluation on RL algorithms.

**Figure 7 sensors-20-05493-f007:**
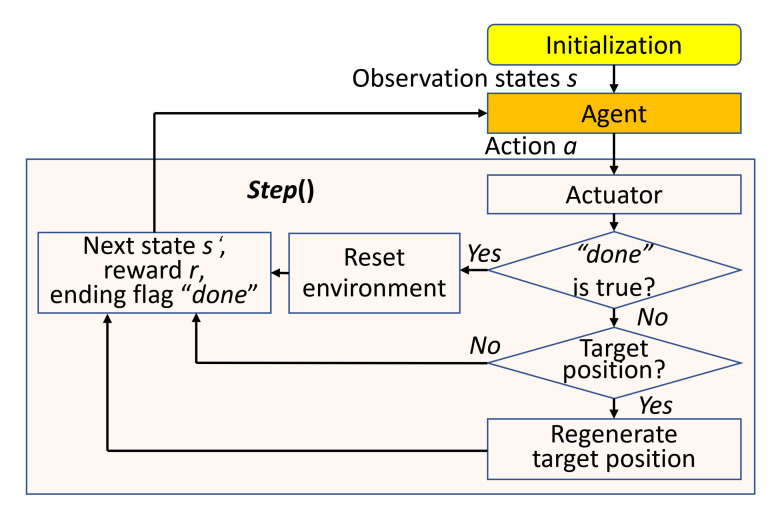
The flowchart of partial DRL environment.

**Figure 8 sensors-20-05493-f008:**
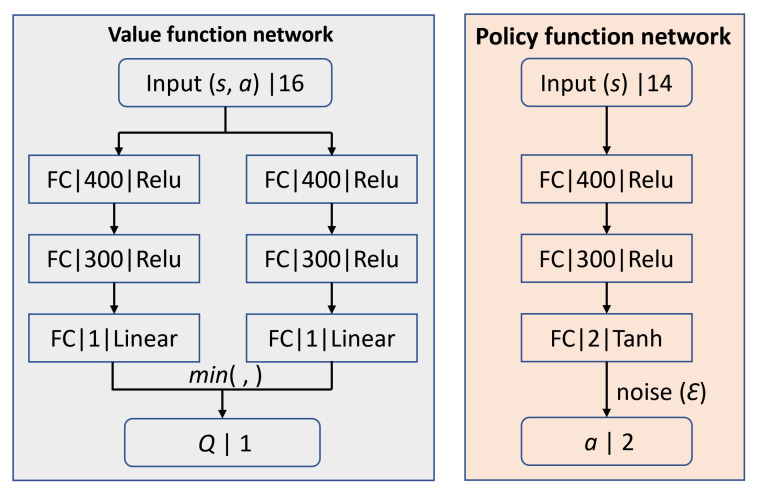
The TD3 network.

**Figure 9 sensors-20-05493-f009:**
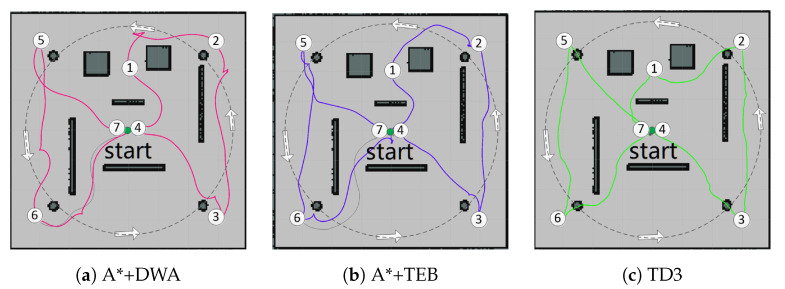
Path planning in small-scale scene.

**Figure 10 sensors-20-05493-f010:**
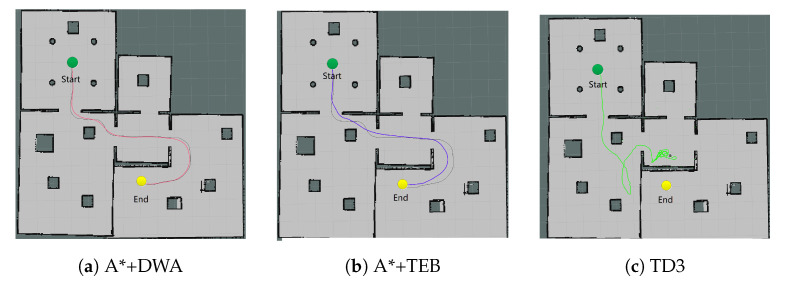
Path planning in large-scale scene.

**Figure 11 sensors-20-05493-f011:**
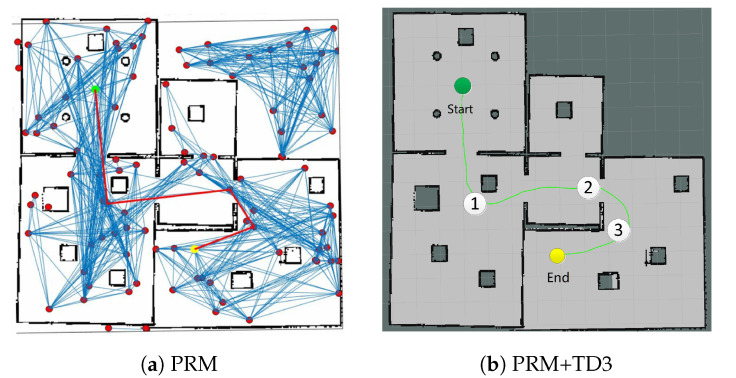
Path planning based on PRM+TD3.

**Table 1 sensors-20-05493-t001:** Evaluation parameters.

No.	Item	Description
1	$d_{ep}$	Path planning map dependence or not (yes/no)
2	$t_{path}$	Duration from start point to end point (s)
3	$t_{step}$	Duration from sensor data input to action command output (s)
4	$l_{path}$	Path length from start point to end point (m)
5	*c*	Percent complete for agent when publish 20 different targets continuously (%)

**Table 2 sensors-20-05493-t002:** Planning indexes in small-scale environment.

No.	Algorithm	$d_{\mathrm{ep}}$	$l_{\mathrm{path}}\left(m\right)$	$t_{\mathrm{step}}\left(s\right)$	$t_{\mathrm{path}}\left(s\right)$	*c*
I	A*+DWA	yes	36.96	0.07∼0.5	256.3	75%
II	A*+TEB	yes	35.11	0.06∼0.7	202.1	85%
III	TD3	no	30.70	0.001∼0.01	171.5	90%

**Table 3 sensors-20-05493-t003:** Planning indexes in small-scale environment.

No.	Algorithm	$d_{\mathrm{ep}}$	$l_{\mathrm{path}}\left(m\right)$	$t_{\mathrm{step}}\left(s\right)$	$t_{\mathrm{path}}\left(s\right)$	*c*
I	A*+DWA	yes	14.54	0.06∼0.3	74.50	95%
II	A*+TEB	yes	13.64	0.05∼0.4	72.10	100%
III	PRM+TD3	yes	15.60	0.001∼0.01	80.24	100%

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
