# Peer review of "Deep Reinforcement Learning for Indoor Mobile Robot Path Planning"

_sensors, 2020, doi:10.3390/s20195493_

Round 1

Reviewer 1 Report

In this paper, a path planning problem for mobile robot in 3D simulation environment was modeled as a Markov Decision Process (MDP). A reinforcement learning algorithm, namely PRM+TD3 (stands for combination of Probabilistic Roadmap with Twin Delayed Deep Deterministic Policy Gradient), was proposed to solve the MDP. In general, this paper is not well prepared and presented. The following comments may help authors to improve the paper:

  1. The Related Work is not well written. Authors claimed the gradual training is novel and PRM+TD3 is novel. The literature review provided in Related Work does not support these claims.
  2. In Line 73-74, “The path planning problem can be modeled as a Partially Observable Markov Decision Process (POMDP), …” In this paper, the path planning problem was modeled as MDP, not POMDP. The listed two references [23,24] used MDP also. In addition, what are state set S and action set A for the proposed problem?
  3. What is the meaning of “network parameters” in the paper?
  4. How to get the reward function presented in Section 3.2? Any reference?
  5. What is the Bellman update for this problem?
  6. The proposed gradual training is a kind of incremental learning, right?
  7. The convergence of the PRM+TD3 should be discussed.
  8. British spellings and American spellings are mixed in the paper, e.g., generalization. In addition, fix the typos, e.g., Turtebot3.

Author Response

1. The Related Work is not well written. Authors claimed the gradual training is novel and PRM+TD3 is novel. The literature review provided in Related Work does not support these claims.

    We thank the reviewer for pointing out the quality and the flaws of the manuscript. We have made significant changes to the language, style, and notations.

    We revised our paper about the Introduction and Related works a lot. To detail our scheme, we rearranged Section 3 and Section 4. We supplemented the traditional path planning approaches such as global planner (Dijkstra, A*, PRM, RRT) , local planner (APF, DWA and TEB) in Section 1 (Introduction). In Section 2 (Related works), we emphasized on research and application including the traditional path planning approaches and DRL-based approaches. Then we proposed our scheme.

2. In Line 73-74, “The path planning problem can be modeled as a Partially Observable Markov Decision Process (POMDP), …” In this paper, the path planning problem was modeled as MDP, not POMDP. The listed two references [23,24] used MDP also. In addition, what are state set S and action set A for the proposed problem?

    Sorry, it is our fault. Exactly, we formulated the path planning problem as MDP, we revised it in Section 3.1. The state set S as the input of DNN, including 10-dimension lidar data, 2-dimension position (angular and distance between the agent and target), 2-dimension velocity (linear and angular). The action set A is the output of DNN, including linear and angular velocity. (details in Section 4.1 and Figure 8).

3. What is the meaning of “network parameters” in the paper?

    The network parameters include the weights W, biases b. the output of neurons is expressed by f(WX+b). We have detailed these parameters in Section 3.2.3.

4. How to get the reward function presented in Section 3.2? Any reference?

    In Section 4.2, we design the reward function based on the literature [16]. In addition, we add an inflation layer. The reference values of reward 200, -150 is determined by experiments.

5. What is the Bellman update for this problem?

    Sorry, in the last manuscript we did not give the details about the update on value and policy functions. In the new manuscript, we added Section 4.4 especially, the Equation (4),(5),(6) to detail the update process based on DRL.  The Equation (4) is adopted by the two Q function to update the value function and use the minimum Q value based on the Bellman Equation to suppress the overestimation problem. The Equation (5) update the Q function by gradient descent to decrease the approximation error, while the Equation (6) update the policy by gradient ascent.

6. The proposed gradual training is a kind of incremental learning, right?

    Exactly! Thank you. The “incremental“ is more understandable. We use “incremental” in the revised manuscript definitely.

7. The convergence of the PRM+TD3 should be discussed.

    We evaluated the convergence of DRL algorithms, including PPO, DDPG, SAC and TD3, based on the 2D environment we build and our path planning tasks in Section 3.4. It can be transfer to the 3D environment easily based on the proposed incremental training mode. In this paper, PRM only supply the way-points for TD3, we validated  it did not change the convergence of our PRM+TD3 planner by experiment.

8. British spellings and American spellings are mixed in the paper, e.g., generalization. In addition, fix the typos, e.g., Turtebot3.

    Sorry about that. We do our best to avoid the typos and spellings in the new manuscript.

Reviewer 2 Report

This paper proposes a gradual training mode with the reward function designed in Equ.(2) to implement the DRL-based path planning for amobile robot. The authors  attempt to transfer the algorithm based on DRL for 2D environment to a simple 3D scenario. Moreover, they also apply a novel path planner, PRM+TD3, to effectively improve the generalisation performance of the model.

This version of the manuscript could not be recommended for publishing in the Sensors due to the following comments.

1.An in-depth introduction of the related work, including their relations to this work, their weakness compared to this work, as well as their concepts behind the work, is also required.

2. I suggest the authors might provide a good table that highlight previous schemes and provide the main characteristics and contributions of the proposed scheme when compared to previous work.

3.The authors are encouraged to explain the concrete contribution of the work. Specially, the authors should clearly explain the Equ. (2) how to work in 2D and 3D environments. Could the authors prove the convergence of the proposed algorithms applied in the DRL model ?

Author Response

1. An in-depth introduction of the related work, including their relations to this work, their weakness compared to this work, as well as their concepts behind the work, is also required.

    We thank the reviewer for pointing out the quality and the flaws of the manuscript. We have made significant changes to the language, style, and notations.

    We revised our paper about the Introduction and Related works a lot. To detail our scheme, we rearranged Section 3 and Section 4.

    We supplemented the traditional path planning approaches such as global planner (Dijkstra, A*, PRM, RRT) , local planner (APF, DWA and TEB) in Section 1 (Introduction). In Section 2 (Related works), we emphasized on research and application including the traditional path planning approaches and DRL-based approaches. Then we proposed our scheme.

2. I suggest the authors might provide a good table that highlight previous schemes and provide the main characteristics and contributions of the proposed scheme when compared to previous work.

    Thank you. In the revised manuscript,  we evaluated the convergence of DRL algorithms, including PPO, DDPG, SAC and TD3 through the figure 6. While the PRM+TD3 planner is compared with A*+DWA, A*+TEB, TD3 through the table 2 and table 3.

3. The authors are encouraged to explain the concrete contribution of the work. Specially, the authors should clearly explain the Equ. (2) how to work in 2D and 3D environments. Could the authors prove the convergence of the proposed algorithms applied in the DRL model ?

    Maybe our little contribution expressed as following: Proposed the incremental training mode (2D to 3D transfer, network parameter transfer), PRM+TD3 planner under different scenes has better generalization.

    In Section 4.2, we design the reward function based on the literature [16]. In addition, we added an inflation layer. The reference values of reward 200, -150 is determined by experiment. In the 2D and 3D environment, we adopted the same network structure, reward function except the state set S (the values of S are calculated in  2D, while the lidar supplies the state values in 3D). The network parameters (W, b) are optimized step by step from 2D to 3D with different scenes.  

    We evaluated the convergence of DRL algorithms, such as PPO, DDPG, SAC and TD3, based on the 2D environment we build and our path planning tasks in Section 3.4. It can be transfer to the 3D environment easily based on the proposed incremental training mode. In this paper, PRM only supply the way-points for TD3, we validated it did not change the convergence of our PRM+TD3 planner by experiment.

Reviewer 3 Report

This is a very interesting study on the application of deep reinforcement learning to the path planning problem for mobile robots, however, I've got some concerns regarding the manuscripts:

1) The introduction should be expanded so that it clearly contrasts the present work against other works in the area, therefore, I literature review is necessary.

2) The introduction also fails to convey the main contributions delivered by this work.

3) Please, proofread the manuscript since I've found some typos and grammar errors.

4) Improve the quality of all figures, especially the ones with simulation/experiment results. The quality of the figures can be improved.

5) The text explaining the flow chart of the gradual training mode, in Figure 5, can be improved. Give more details on the algorithm steps and how it is or can be practically implemented.

Author Response

1. The introduction should be expanded so that it clearly contrasts the present work against other works in the area, therefore, I literature review is necessary.

    We thank the reviewer for pointing out the quality and the flaws of the manuscript. We have made significant changes to the language, style, and notations.

    We revised our paper about the Introduction and Related works a lot. To detail our scheme, we rearranged Section 3 and Section 4.

    We supplemented the traditional path planning approaches such as global planner (Dijkstra, A*, PRM, RRT) , local planner (APF, DWA and TEB) in Section 1 (Introduction). In Section 2 (Related works), we emphasized on research and application including the traditional path planning approaches and DRL-based approaches. Then we proposed our scheme.

2. The introduction also fails to convey the main contributions delivered by this work.

    We supplemented the traditional path planning approaches such as global planner (Dijkstra, A*, PRM, RRT) , local planner (APF, DWA and TEB) in Section 1 (Introduction). In Section 2 (Related works), we emphasized on research and application including the traditional path planning approaches and DRL-based approaches. Then we proposed our scheme.

3.  Please, proofread the manuscript since I've found some typos and grammar errors.

    Sorry about that. We do our best to avoid the typos and grammar error in the revised manuscript.

4. Improve the quality of all figures, especially the ones with simulation/experiment results. The quality of the figures can be improved.

    Thank you! We update all the figures. To detail our scheme, we added the figure 7 and 8. Figure 7 denotes the process to build the DRL environment used in this paper. Figure 8 includes the value function network (two-Q) and policy function network, in where each layer is consisted of connection type, neurons and activation function except the input layer. TD3 fuses the value and policy functions, which adopt the similar DNN.

5. The text explaining the flow chart of the gradual training mode, in Figure 5, can be improved. Give more details on the algorithm steps and how it is or can be practically implemented.

    Thank you! We revised this figure (Figure 4, in the revised manuscript) and explained the incremental training process in details in Section 3.2, including environment/network structure/parameter transfer, and the learning process.

Reviewer 4 Report

The authors propose a Deep Reinforcement Learning based model for robot path planning in indoor environments. Although the paper is well-written, and the problem and the proposed method are interesting, I believe the paper could be further improved in the following areas.

  1. The paper's most critical problem is the lack of comparison to the state-of-the-art DRL-based methods in path planning. Two of these methods are mentioned at the end of the review as examples [1, 2]. It could be more interesting to compare your approach to those methods and draw a distinction between the proposed model with them.
  2. How have the values in Eq(2) been chosen? Is there any proof or intuition behind the values for the rewards?
  3. You have not mentioned the architecture of the neural network and the related configurations in the paper. I believe it could help the readers to understand the method. It can also improve the reproducibility of your research.
  4. In section 5.2, it is mentioned that in the presented model, collisions occur in the starting phase when the obstacles are too close to the robot. Are there any experiments reflecting this claim? Why does this happen? I am particularly interested in seeing the results of state-of-the-art models and your model in more crowded environments.

Given all these comments, I believe the paper needs to go under a major revision to be accepted.

[1] Zeng, J.; Qin, L.; Hu, Y.; Yin, Q.; Hu, C. Integrating a Path Planner and an Adaptive Motion Controller for Navigation in Dynamic Environments. Appl. Sci. 2019, 9, 1384.

[2] Sanchez-Lopez, J.L., Wang, M., Olivares-Mendez, M.A. et al. A Real-Time 3D Path Planning Solution for Collision-Free Navigation of Multirotor Aerial Robots in Dynamic Environments. J Intell Robot Syst 93, 33–53 (2019). https://doi.org/10.1007/s10846-018-0809-5

Author Response

1.The paper's most critical problem is the lack of comparison to the state-of-the-art DRL-based methods in path planning. Two of these methods are mentioned at the end of the review as examples [1, 2]. It could be more interesting to compare your approach to those methods and draw a distinction between the proposed model with them.

    Thank you. In the revised manuscript, we analyzed the literatures [1][2] you advised. We added Section 3.4 and evaluated the convergence of RL algorithms, such as PPO, DDPG, SAC and TD3, based on the 2D environment we build and our path planning tasks in Section 3.4. It can be transfer to the 3D environment easily based on the proposed incremental training mode.  We supplemented and revised the experiment and analysis in Section 5.   Thank you very much. The methods based on [1][2] are worth following for our next work.

2. How have the values in Eq(2) been chosen? Is there any proof or intuition behind the values for the rewards?

    In Section 4.2, we design the reward function based on the literature [16]. In addition, we add an inflation layer. The reference values of reward 200, -150 is determined by experiments.

3. You have not mentioned the architecture of the neural network and the related configurations in the paper. I believe it could help the readers to understand the method. It can also improve the reproducibility of your research.

   Thank your for your advises. We added Section 4.4 especially, the figure 8 and the Equation (4),(5),(6) to detail the update process based on DRL. Figure 8 includes the value function network (two-Q) and policy function network, in where each layer is consisted of connection type, neurons and activation function except the input layer. TD3 fuses the value and policy functions, which adopt the similar DNN.  The Equation (4) is adopted by the two Q function to update the value function and use the minimum Q value based on the Bellman Equation to suppress the overestimation problem. The Equation (5) update the Q function by gradient descent to decrease the approximation error, while the Equation (6) update the policy by gradient ascent. The state set S as the input of DNN, including 10-dimension lidar data, 2-dimension position (angular and distance between the agent and target), 2-dimension velocity (linear and angular). The action set A is the output of DNN, including linear and angular velocity. (details in Section 4.1 and Figure 8).

4. In section 5.2, it is mentioned that in the presented model, collisions occur in the starting phase when the obstacles are too close to the robot. Are there any experiments reflecting this claim? Why does this happen? I am particularly interested in seeing the results of state-of-the-art models and your model in more crowded environments.

    With the inherent mechanical characteristics, the collision seldom occurs at the starting phase, when the agent and obstacles are very close. Because of the embedded physical engine ODE in Gazebo, the acceleration and velocity of the agent are smaller than the mobile obstacles, it is consistent with the practice and the collision will disappear at the normal running phase.

    Thank you for your advises. We added Section 3.4 and evaluated the convergence of RL algorithms, such as PPO, DDPG, SAC and TD3, based on the 2D environment we build and our path planning tasks in Section 3.4. It can be transfer to the 3D environment easily based on the proposed incremental training mode.  We supplemented and revised the experiment and analysis in Section 5.  

Round 2

Reviewer 1 Report

In this revised manuscript, authors carefully addressed most of reviewer's concerns and comments. The manuscript quality has been significantly improved. The following comments may help authors to further improve the paper:

  1. Define the abbreviation the first time you use it in the text. e.g. PRM and TD3 in the Abstract, DRL in the Introduction;
  2. In line 23, instead of using "former [6], authors may use Levine et al. [6];
  3. Directly use author's last name to cite the reference. e.g., change "Pfeiffer, M., et al., [21]" to "Pfeiffer et al. [21]; 
  4. Incremental training is not new, however, no related research has been mentioned and compared in the Related Work to justify the novelty. In addition, change "Related Works" to "Related Work";
  5. Correct typo, e.g., in line 63, RM+TD3.

Author Response

In this revised manuscript, authors carefully addressed most of reviewer's concerns and comments. The manuscript quality has been significantly improved. The following comments may help authors to further improve the paper:

1.Define the abbreviation the first time you use it in the text. e.g. PRM and TD3 in the Abstract, DRL in the Introduction;

     We thank you for your advises. It’s our misunderstanding about the location of the abbreviation should be. We have defined all the abbreviations in the first time we use.

2. In line 23, instead of using "former [6], authors may use Levine et al. [6];

      Thanks for your very kindly review.  Here, the former indicates global planner, which may be easy to confuse. We have revised it.

3. Directly use author's last name to cite the reference. e.g., change "Pfeiffer, M., et al., [21]" to "Pfeiffer et al. [21]; 

    We appreciate it! We think we can avoid this de-normalized writing after this.

4. Incremental training is not new, however, no related research has been mentioned and compared in the Related Work to justify the novelty. In addition, change "Related Works" to "Related Work";

     Yes. It’s not new. In the related work we discuss the related path planner, such as Dijkastra and DWA +CNN[20],  PRM+RL[24], JPS-IA3C[26], and so on. We did not find papers on PRM+TD3 planner (we adopted) as we can search. May be our search range is not too wide at present. However, we will do further study and research on robot path planner. Thank you for your helpful advises. 

5. Correct typo, e.g., in line 63, RM+TD3.

    Thank you! It’s our negligence. Have revised it.

Reviewer 2 Report

This revision can be accepted as a paper for Sensors.

Author Response

Thank you very much for your helpful advises.

We revised our paper about typos, abbreviations, related work [6] in line 23-24. Directly use author's last name to cite the references.

Best regards!

Reviewer 4 Report

The response from the authors appears to have addressed the concerns raised by this reviewer. The figures, the content and the arguments given by the authors seems to be satisfactory.